



# Orientation selective grain sublimation-deposition in snow under temperature gradient metamorphism observed with Diffraction Contrast Tomography

Rémi Granger[*,1,2], Frédéric Flin[2], Wolfgang Ludwig[3], Ismail Hammad[3], and Christian Geindreau[1]

[1]Univ. Grenoble Alpes, CNRS, Grenoble INP[†], 3SR, F-38000 Grenoble, France
[2]Univ. Grenoble Alpes, Université de Toulouse, Météo-France, CNRS, CNRM, Centre d'Etudes de la Neige, 38000 Grenoble
[3]ESRF, BP 220 38043 Grenoble Cedex, France

**Abstract.** In this study on temperature gradient metamorphism in snow, we investigate the hypothesis that there exists a favorable crystalline orientation relative to the temperature gradient, giving rise to a faster formation of crystallographic facets. We applied in-situ time-lapse Diffraction Contrast Tomography on a snow sample with a density of 476 $\mathrm{kg\,m^{-3}}$ subject to a temperature gradient of 52 $\mathrm{^\circ C\,m^{-1}}$ at mean temperatures in the range between -4.1 °C and -2.1 °C for three days. The

5 orientations of about 900 grains along with their microstructural evolution are followed over time. Faceted crystals appear during the evolution and from the analysis of the material fluxes, we indeed observe higher sublimation-deposition rate for grains with their c-axis in the horizontal plane at the beginning of the metamorphism. This remains the case up to the end of the experiment for what concerns sublimation while the differences vanish for deposition. That latter observation is explained in terms of geometrical interactions between grains.

## 10 1 Introduction

The snow cover consists of a porous medium made of ice crystals, air and sometimes liquid water. The microstructure of this material is in constant evolution and the original fresh snow is transformed under the action of various mechanisms called snow metamorphism. In particular, a temperature gradient (TG) in a snow layer implies a gradient of water vapor density, which creates a vapor flux in the pore space. Under these circumstances, ice crystal interfaces may be significantly out of

15 equilibrium with the vapor phase if the macroscopic temperature gradient is higher than 10-20 $\mathrm{^\circ C\,m^{-1}}$. Of course, these values should just be considered as orders of magnitude, since the local temperature gradient is strongly depending on the geometrical configuration at the grain scale, as e.g. pointed out by Hammonds et al. (2015) and Hammonds and Baker (2016). Growth and decay under TG conditions depend strongly on the kinetics of the deposition and sublimation processes on different crystallographic facets, and as a consequence, kinetic forms (faceted crystals and depth hoar) of snow crystals appear (Colbeck,

20 1983). An important aspect of temperature gradient metamorphism is that the grain size increases while the density remains constant (Marbouty, 1980; Schneebeli and Sokratov, 2004; Srivastava et al., 2010; Pinzer et al., 2012; Calonne, 2014). Indeed,

---

[†]Institute of Engineering Univ. Grenoble Alpes
[*]e-mail: remi.granger@3sr-grenoble.fr



due to the local temperature field, some parts of the microstructure grow at the expense of others (Akitaya, 1974; Pinzer et al., 2012), ensuring an almost null net flux of ice far from the system boundaries. In addition, the kinetic coefficient is not equal on the prismatic and basal faces of ice (see e.g. Yokoyama and Kuroda, 1990; Libbrecht, 2005; Furukawa, 2015). In consequence, growth and decay of a ice crystal in a vapor gradient depend on the orientation of the crystal relative to the thermal (and also

water vapor saturation pressure) gradient, with higher sublimation and deposition rates when the more active faces efficiently catch and release the flux.

In that context, Adams and Miller (2003) and later Miller and Adams (2009) conjectured that, in temperature gradient metamorphism, favorably-oriented crystals might grow at the expense of others as they grow more quickly towards the common vapor source. Because of that selective growth of crystals, the distribution of crystallographic orientation, called the fabric, is

expected to evolve towards an anisotropic fabric. However, few studies report data on the subject. Takahashi and Fujino (1976) and de Quervain (1983) measured crystal orientation but the data were too scarce to draw conclusions.

More recently, Riche et al. (2013) measured the fabric evolution over natural and sieved samples under a $50\,°\text{C}\,\text{m}^{-1}$ temperature gradient for 12.5 weeks, at a mean temperature of -20 °C. They used the Automatic Ice Texture Analyser (AITA) (Wilson et al., 2007) that enables measurement of crystallographic orientations of thin sections. They observed an evolution of the

fabric from strong cluster-type (with c-axis of ice crystals mainly in one direction) to weak girdle-type (with c-axis mainly in a plane) for the natural samples, while no evolution for the denser sieved sample was observed. Calonne et al. (2017) analysed snow-firn profiles from Antarctica in terms of i) microstructure using microcomputed tomography and ii) fabric, using AITA. They showed that the snow fabric is correlated with its microstructure; the most isotropic fabric being found in the densest snow and smallest specific surface area (SSA). A similar study applied to EastGRIP data (Greenland) revealed some strong

cluster-type textures at shallow depth (Montagnat et al., 2020). All of these recent studies suggest that metamorphism affects the fabric.

While AITA is an efficient instrument to obtain statistical data on fabric and on different samples, it requires thin sectioning of the sample. Consequently, it is not possible to follow the time evolution of individual grains and observe the role of crystal orientation at the grain scale. To bring such complementary understanding into the picture, we present here an experiment of

temperature gradient metamorphism, monitored over a duration of 3 days by conventional tomography and Diffraction Contrast Tomography (DCT). DCT is a near-field X-ray diffraction imaging modality exploiting the Bragg diffraction signals created by the individual crystals in the illuminated sample volume. By measuring the position and shape of the associated diffraction spots, it is possible to reconstruct the crystallographic orientation and shape of each of the grains (Ludwig et al., 2009; Reischig et al., 2013). This technique allows for non-destructive in-situ observations of extended sample volumes and has already been

used for snow to follow deformation under compression (Rolland du Roscoat et al., 2011).

The paper is structured as follows: in the first section, we describe the sample preparation, the experimental setup and the principal image processing steps. In the second section, we present a time series of the evolving sample microstructure as well as the characterization and analysis of the relation between grain orientation and mass fluxes. Finally, we discuss the observed trends, the representativity of the experiment, as well as the main technical difficulties met with the imaging technique.



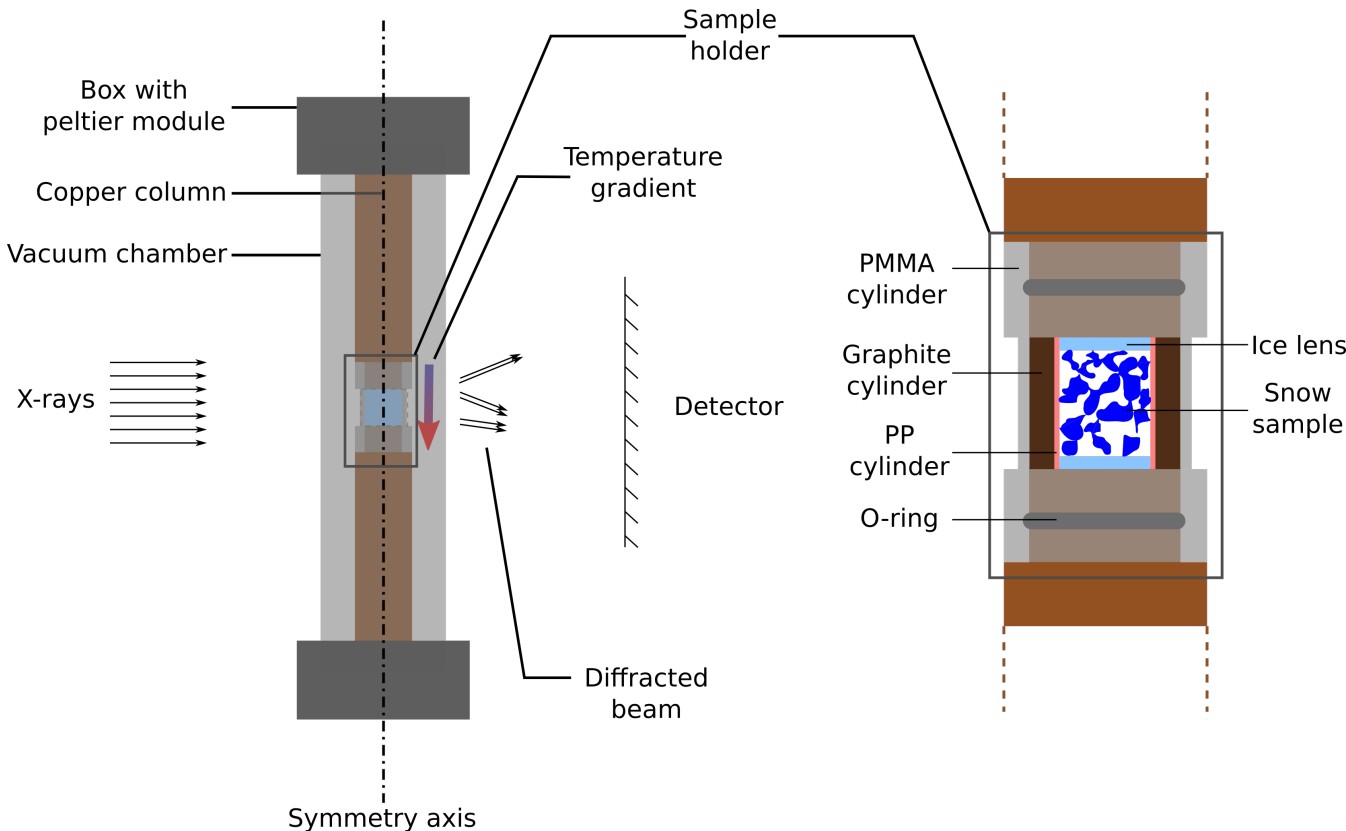

**Figure 1.** Scheme of the cryogenic cell CellDyM2, the DCT-compliant version of CellDyM (Calonne et al., 2015).

## 2 Methods

### 2.1 Sample and sample environment

To impose controlled temperature conditions on the sample, we used CellDyM2 (see Fig. 1), a modified version of the cryogenic cell developed by Calonne et al. (2015): as for CellDyM, the sample is inserted in a sample holder placed between two copper columns, at the extremities of which temperature is controlled using Peltier modules. The sample holder and the copper columns are placed in a vacuum chamber for thermal insulation from room temperature. However, CellDyM2 was specifically adapted to DCT experiments. Indeed, the original aluminium sample holder, with typically large crystalline grains, can create parasitic diffraction spots and thus has been replaced by a system of three components:

- a 1 mm thick polymethyl methacrylate (PMMA) holder of inner diameter 10 mm,

- a 3 mm thick graphite cylinder of inner diameter 4 mm,

- a thin polypropylene (PP) tube inside the graphite cylinder.



The PMMA container, combined with two rubber O-rings, provides a vacuum-tight assembly. The graphite, of thermal conductivity 95 W m$^{-1}$ K$^{-1}$ and grain size below $\approx 5$ µm, ensures a good thermal conductivity between the two copper columns and, thus, a good control of the temperature gradient without problematic parasitic diffraction spots. Finally, the PP cylinder avoids the diffusion of water vapor inside the porous graphite and preserves the sample from potential contamination by carbon

dust. The vacuum in the chamber is produced by a system of two pumps: a primary scroll pump (Anest-Iwata ISP-90) produces a pressure of about 500 Pa, which permits to a turbomolecular pump (Pfeiffer Vacuum HiPace 30), directly mounted on the turntable, to reach the final pressure of 0.009 Pa.

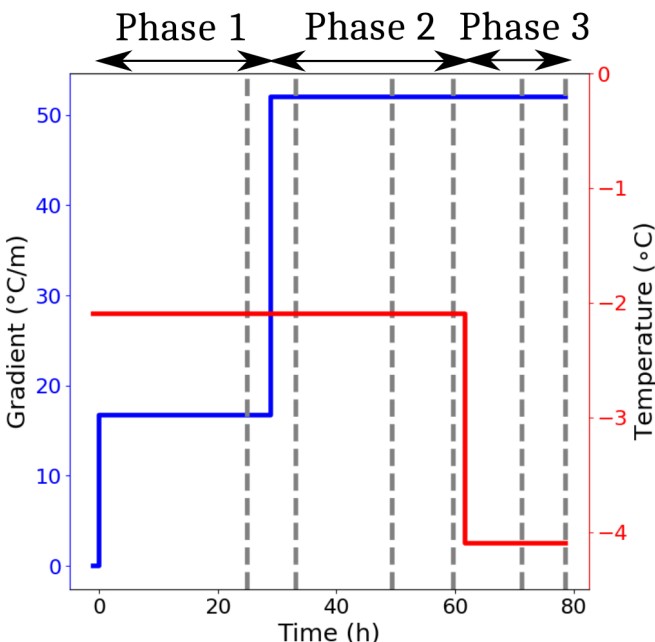

**Figure 2.** Evolution of the mean temperature and downward temperature gradient imposed on the sample.

The cylindrical sample volume with 4 mm diameter and 4 mm height contained rounded grain snow with a density of $\approx 450$ kg m$^{-3}$ and was submitted to a downward temperature gradient. To ensure constant vapor flux boundary conditions to the snow

sample, the top and bottom surfaces of the whole 10 mm high cylinder consisted of 3 mm thick ice lenses, serving as a matter sink or source, respectively. In order to minimize parasitic diffraction spots, these lenses were machined from monocrystals, using a specific core drill. The sample was prepared in a cold room at CEN. The original fresh snow was picked up at the upper station of La Grave cable car and stored at -20 °C for 7 months before it was metamorphised at -5 °C in equilibrium conditions for 3 days. A specimen was then cored in a homogeneous zone, and assembled on the bottom copper column with the bottom

ice lens. The top ice lens was then stuck to the top copper column, which was hence placed at the upper part of the sample. The assembly was then stored in a specific copper handling tube, preserving the sample from possible temperature gradients during storage. The whole assembly was then transported to ESRF within a carriable freezer and inserted in the cell at -20 °C.





As soon as the cell was closed, pumping the air out from the vacuum chamber started, leading to a pressure of 0.046 Pa in less than 27 min. It took $\approx$ 7 h to reach the stable pressure of 0.009 Pa. Then, the regulation of both of the Peltier modules was progressively set to -2.10 °C. This value was reached in about 1 hour and the sample stayed in equilibrium conditions for 21 additional hours. Then, the proper temperature gradient experiment began: this corresponds to the instant $t = 0$ in this paper.

The thermal conditions applied to the snow sample during the experiment are shown in Figure 2. The experiment consists in three phases that we will now refer to as phase 1, phase 2 and phase 3. Mean temperature and temperature gradients were respectively set to -2.09 °C and 17 °C m$^{-1}$ during phase 1, -2.09 °C and 52 °C m$^{-1}$ during phase 2, and -4.07 °C and 52 °C m$^{-1}$ during phase 3. These conditions were determined by performing numerical simulations of heat transfer in the cell using COMSOL Multiphysics, as presented by Calonne et al. (2015). In all these simulations, the temperatures imposed by the
Peltier modules were used as inputs in the numerical model.

## 2.2   Diffraction Contrast Tomography: Experimental Setup

The experiment was performed at beamline ID19 of the European Synchrotron Radiation Facility (ESRF) in a period of 4 bunch operation of the storage ring. A beam energy of 40 keV was selected using a Silicon double crystal monochromator in Laue-Laue setting. Detection was performed using detector system consisting of a 200 µm LuAG screen coupled via visible
light optics to a sCMOS camera (2048 x 2048 pixels), resulting in an effective pixel size of 6.5 µm and a field of view of 13 mm. The distance between the rotation axis and the detector was 60 mm.

    DCT scans consisted of 7200 projection images, acquired during continuous rotation of the sample over 360 degree with an angular step size of 0.05 ° and exposure time of 50 ms, resulting in a total scan time of about 1 hour. In order to reduce background, the area of the scintillation screen illuminated by the direct beam was covered by a 5x5 mm$^2$ beamstop, made from
0.5 mm lead sheet. Absorption scans, acquired with the same detector, shifted laterally by 5 mm, consisted of 900 images with angular step size of 0.4 ° and 0.05 exposure time ($\approx$ 10 min scan time). DCT and absorption scans were performed alternatively so that there was systematically an absorption scan performed in the 20 minutes before a DCT scan.

    Reconstruction was performed following the workflow detailed in Reischig et al. (2013). A brief outline of the processing route is summarized below:

1. Preprocessing: distortion and background correction are applied to the image stack.

    2. Segmentation of diffraction spots: double soft thresholding.

    3. Matching Friedel pairs: when a crystalline plane $(hkl)$ meets the Bragg's condition and leads to a diffraction spot for a position $\omega$ of the rotation stage, then the Bragg condition is met again at $\omega + 180°$ for the $(\bar{h}\bar{k}\bar{l})$ plane and the two associated diffraction spots are called a Friedel pair, which are identified automatically using criteria based on the
position, intensity and shape of the segmented diffraction spots. Over 360°, a crystallographic plane in one grain can meet the Bragg conditions up to four times and so gives rise to two Friedel pairs. A Friedel pair allows inferring the trajectory of the diffracted beam and hence to derive the direction of the $(hkl)$ plane normal.





4. Indexing grains: the position and orientation of grains are determined by identifying consistent Friedel pairs of diffraction spots corresponding to the same grain.

5. Collection of additional diffraction spots by forward simulation and final selection of diffraction spots.

6. Reconstruction of 3D grain shape using the selected diffraction spots for each grain using the SIRT algorithm (Palenstijn et al., 2013, 2015).

7. Assembly of the 3D grain volume: the individually reconstructed grain volumes are assembled into the common sample volume.

8. Masked dilation: morphological dilation of the reconstructed grain map constrained to the domain occupied by ice, as determined from segmentation of the sample volume obtained from the absorption contrast scans.

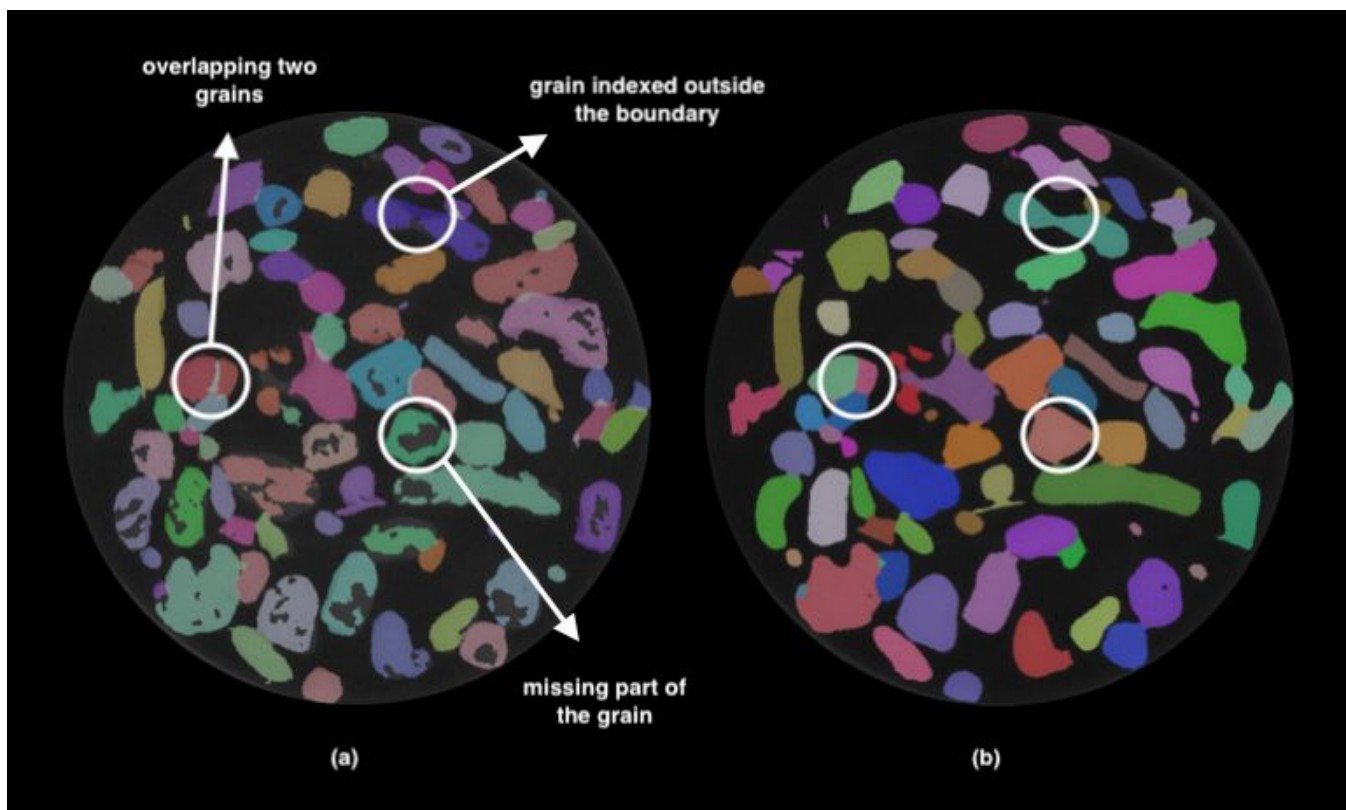

**Figure 3.** Effect of step 8 to improve the reconstruction quality: before (a) and after (b) correction.

The last step of masked dilation improves the accuracy of shape reconstruction in porous (and/or multiphase) media, by enforcing fidelity of the grain shape at external grain interfaces (or phase boundaries). Ice crystals in snow are nearly perfect and in our case of dimensions comparable or bigger than the Pendellösungs lengths: as a consequence, dynamic (multiple)



diffraction leads to noticeable deviations of the local diffracted intensity from the idealized, linear projection model (kinematic diffraction conditions) upon which our implementation of the reconstruction algorithm is based. These non-linearities and remaining diffraction spot overlaps, not filtered out in previous consistency checks, give rise to inconsistent grain shape reconstructions, i.e. streaks and modulations of the reconstructed local scattering power inside the grain volumes. Since the final grain map is assembled from a segmentation of these individual gray-scale grain volumes, these artefacts can result in (i) inaccurate contours created by overlap between neighboring grains, (ii) inaccurate contours created by overlap with the pore space of the material and (iii) internal holes or holes connected to the contour of the segmented grains.

Since the localization of ice phase can be easily obtained from the absorption scans, the masked dilation can be used to correct part of the inaccuracies introduced by dynamical diffraction (for instance internal holes). Figure 3 shows the effect of this correction on a horizontal slice of the sample. While this procedure enables recovery of the correct information for a large fraction of the volume elements (voxels), one has to be aware that it may also lead to the assignment of wrong orientations for voxels in the vicinity of the grain boundaries. This point will be discussed in section 4 in the light of the obtained results.

## 2.3 Data processing

Reconstruction of the absorption scans was performed using filtered back projection (FBP) algorithm and ring artefacts were corrected using an in-house algorithm.

Because some movements of the stages and detector may occur during data acquisition, it is necessary to register the time series of the reconstructed volumes. To simplify that step, reference grooves have been previously made on the exterior side of the PP cylinder following both the axis and the circumference. First, all absorption volumes were registered on a reference volume (taken at t = 27 h 32 min, at the begining of phase 2). Registration was performed using the SimpleITK library and consists of:

- smoothing the image with a Gaussian filter with kernel size of 2 voxels,

- finding the optimal translation in the horizontal plane,

- finding the optimal vertical translation,

- applying the total transform to the not smoothed reconstructed volumes with linear interpolation.

The optima were defined by maximizing the correlation computed in a hand selected ROI focusing on the grooves. Registered volumes were then segmented using an energy based segmentation algorithm (Hagenmuller et al., 2013).

Some grains may be missing during the DCT reconstruction because the diffracted beam may not hit the detector or because the quality and quantity of grouped diffraction spots may not be sufficient for reconstruction. Correcting such data requires manual monitoring and processing throughout the sample volume. For that reason, only a selected subset of the DCT scans was processed in that study. Times at which a processed volume is available are represented by grey dotted lines in Figure 2. The correction consists in comparing the reconstructed absorption and DCT scans to detect missing grains, and by comparing the former and latter volumes in the time series, to get the orientation of the grains which remain fix during the experiment,



as no rearrangment occurs. After that, because DCT scans were always performed shortly after the absorption scans, they were directly registered on the closest absorption volume by finding the optimal translation in space that maximize correlation between the two volumes, without ROI selection.

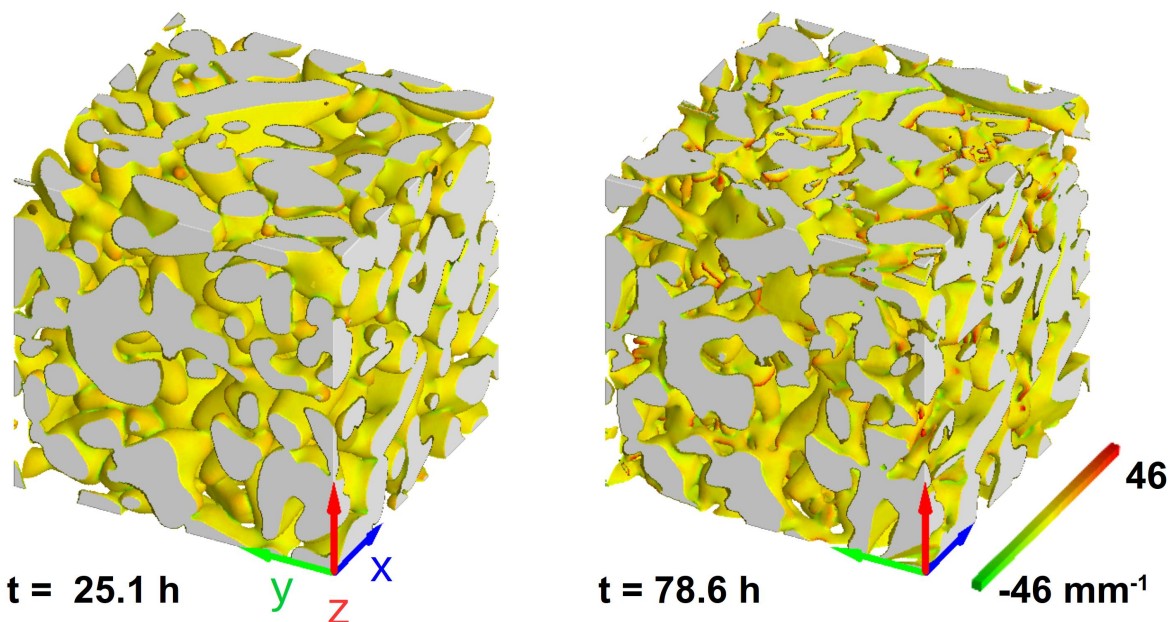

**Figure 4.** Time evolution of the microstructure in a cube of 2.34 mm edge length cropped out of the center of the specimen. The interfaces are couloured based on the local mean curvature.

## 3   Results

### 3.1   Image series

Figure 4 shows images of the microstructure at $t = 25.1\,\mathrm{h}$ and $t = 78.6\,\mathrm{h}$ of a cubic subsample of edge length 2.34 mm in the center of the sample. The colours represent the mean curvature computed with an algorithm developed by Flin et al. (2004, 2005) - see also (Calonne et al., 2014), and ranges from -46 to 46 $\mathrm{mm}^{-1}$. First, we can see that the initial microstructure of the snow consists of small well rounded grains. The global evolution is small as full recrystallization of the grains is not observable. Newetheless, all the typical features of temperature gradient metamorphism can be observed, e.g.:

– a strong faceting of the downward oriented surfaces, as revealed by the grain geometry and the associated curvature field,

– a global migration of the ice microstructure towards the warmer side of the sample.





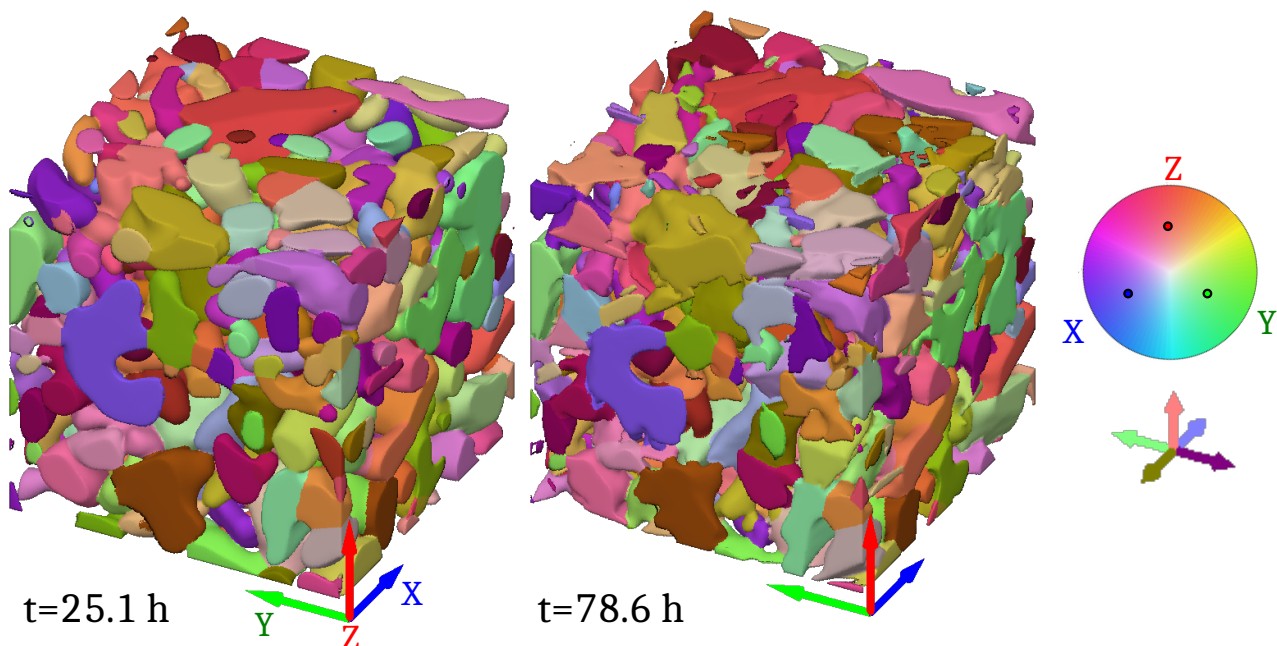

**Figure 5.** Grain map evolution. Each colour represents a single crystalline orientation. The meaning of the colours is given by the colour wheel, repesenting orientations in a pole figure centered on a vector $\mathbf{X} + \mathbf{Y} + \mathbf{Z}$. The 5 axes under the colour wheel show the colour corresponding to the main directions associated to $\mathbf{X}, \mathbf{Y}$ and $\mathbf{Z}$.

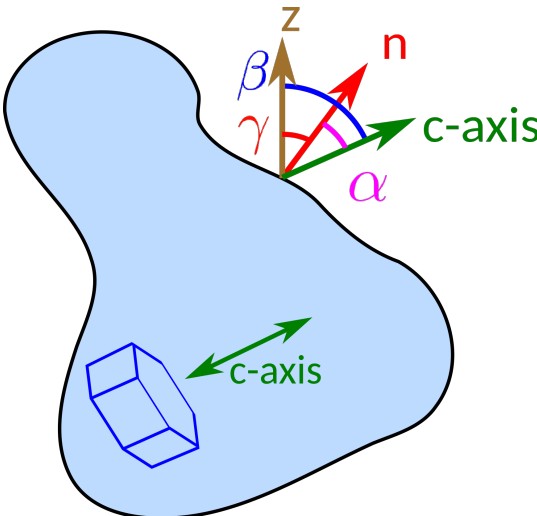

**Figure 6.** Parametrization of the different vectors and angles: $\mathbf{z}$ is the vertical unit vector, $\mathbf{n}$ is the local normal outgoing from the ice and $\mathbf{c}$ is a vector with positive vertical component representing the c-axis direction.





In Figure 5, we present the result of the DCT acquisition obtained after the processing route presented in section 2.3. The orientation of each grain is characterized by a unit vector **c** which, by convention, has positive vertical component. On the figure, the direction of c-axis is represented by a colour given by the colour wheel. From a grain to grain close image comparison, we can see that no grain rearrangement can be observed (no c-axis rotation), but just a modification of the grain

5    shapes and boundaries, which is consistent with the current knowledge of TG snow metamorphism.

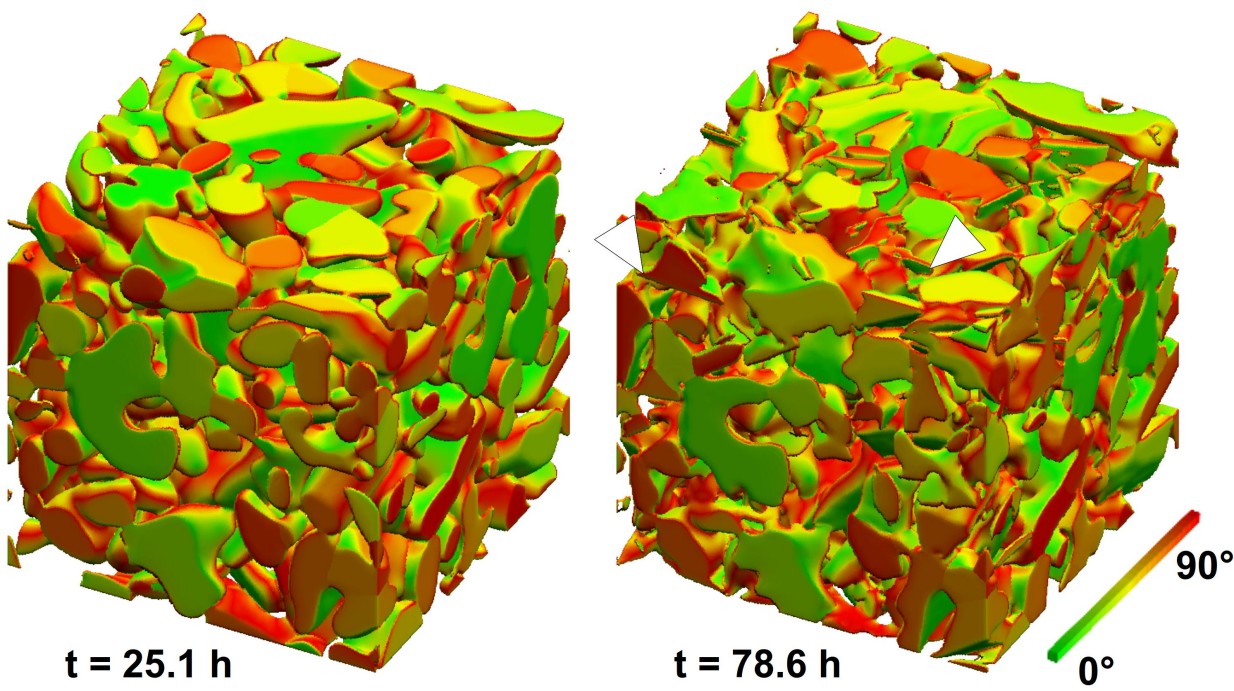

**Figure 7.** Time evolution of the facet orientation. The colours represent the angle between the c-axis obtained from the DCT information and the normal to the interface: green represents basal faces, and red corresponds to cases where the surface is parrallel to the c-axis. White arrowheads point out plate-like crystals formed at the end of the experiment, where consistency between orientation and morphology can be checked.

For further analysis, we introduce the parametrization represented in Figure 6. At a given point of the surface, three vectors have importance in the physics of the metamorphism: (i) the vertical direction, given by **z** pointing upward, which is the direction of the global macroscopic vapor flux, (ii) the normal at the interface **n**, taken here as pointing into the pore space, and (iii) the c-axis given by **c**. We can then define three angles: $\alpha = (\mathbf{c}, \mathbf{n})$, $\beta = (\mathbf{z}, \mathbf{c})$ and $\gamma = (\mathbf{z}, \mathbf{n})$. Because we are only

10    interested in the direction of **c**, values range is [0,90°] for $\alpha$ and $\beta$, and [0,180°] for $\gamma$.

Figure 7 represents the microstructure with the interface coloured by the angle $\alpha$ so that the basal faces ($\alpha = 0°$) are green and the prismatic faces ($\alpha = 90°$) are red. By looking particularly at the plates formed during the metamorphism, we can observe that the orientation of the c-axis is consistent with their morphology.

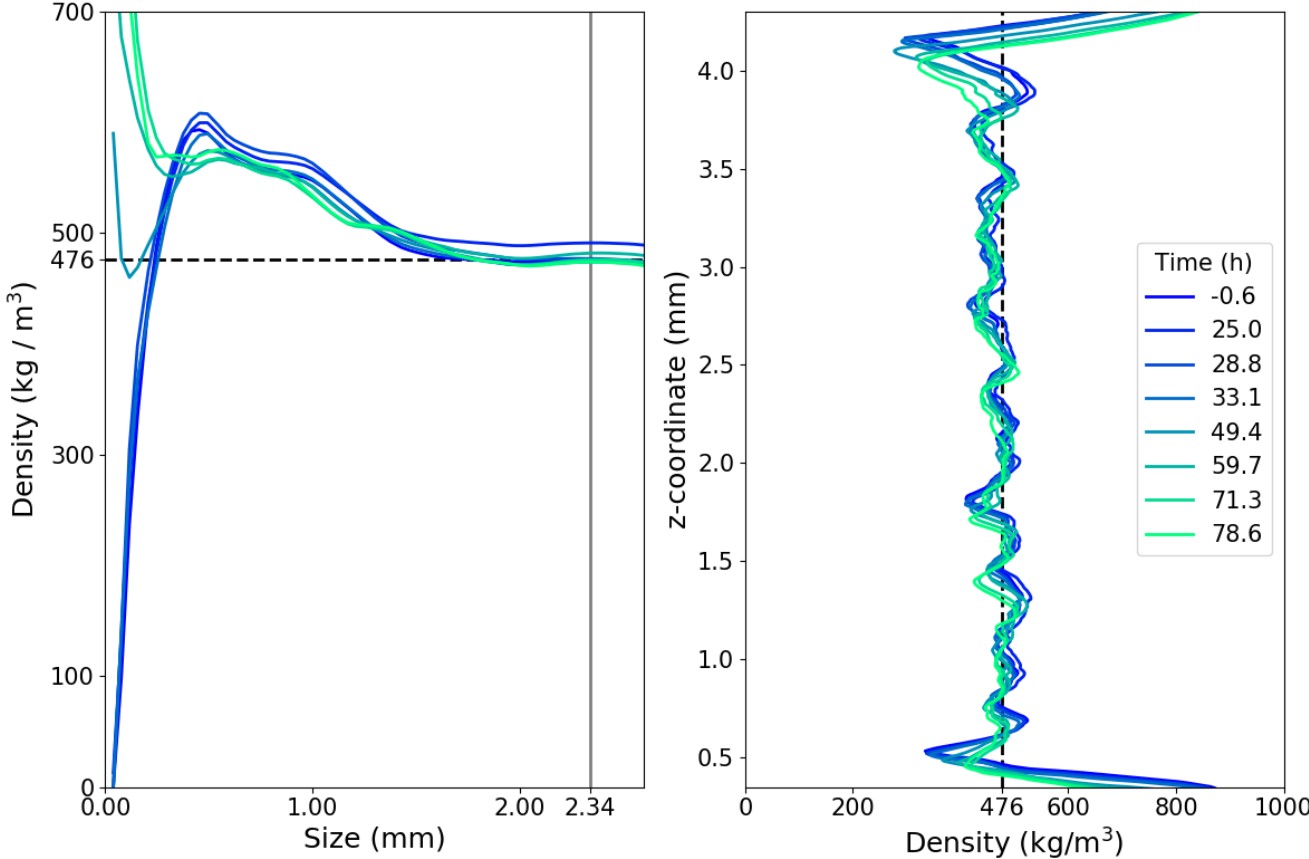

**Figure 8.** Density measurements. a) Representative Elementary Volume estimation for density, computed from centered cubic subvolumes of variable sizes. b) Vertical variation of density computed on centered discs of 1 voxel height and 3.3 mm in diameter.

## 3.2 Density

Representative Elementary Volume (REV) analysis for density was performed on absorption volumes at different times of the experiment, using cubic centered subvolumes of increasing sizes. The results are depicted in Figure 8.a, and show that the density reaches a constant value for cubes of edge about 2.34 mm. In order to complement the characterization of the homogeneity, we computed the density of horizontal discs of the sample, centered on the vertical axis of the sample with a diameter of 3.3 mm (so that it corresponds to the cubic REV of 2.34 mm edge) and one voxel (6.5 μm) high. This enables one to obtain a vertical density profile. The results are shown in Figure 8b.

The high values on top and bottom correspond to ice lenses ($\rho_{ice} = 917$ kg m$^{-3}$). Height and time fluctuations are visible, but the sample is overall homogeneous in the vertical direction. We also computed the density for each absorption volume on

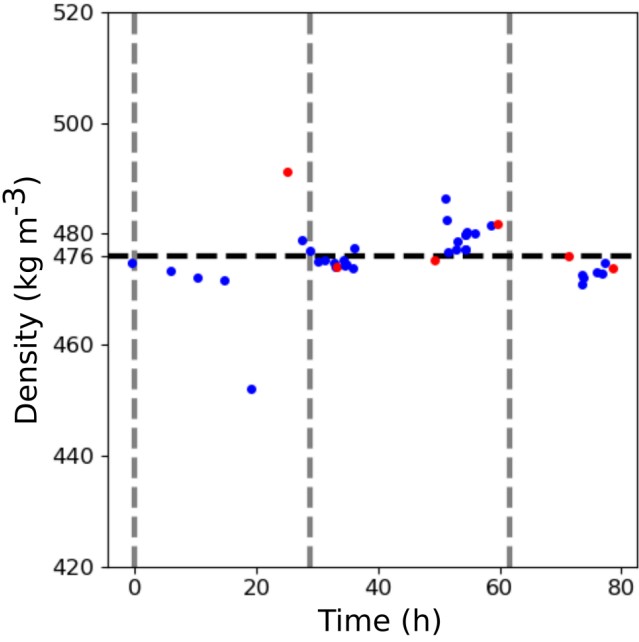

**Figure 9.** Density evolution: grey vertical lines represent the times of change of temperature conditions. Red dots correspond to times for which DCT information is available. The horizontal black dashed line represents mean density, which stays around $(476 \pm 5\%)$ kg m$^{-3}$.

a cubic REV of size 2.34 mm. The result is shown in Figure 9. We observe a constant density throughout the experiment at the initial value of $(476 \pm 5\%)$ kg m$^{-3}$.

### 3.3 Specific Surface Area

First, REV for SSA where computed on centered cubic subvolumes of increasing sizes, at various times during the experiment. SSA computations were performed using an improved version of the VP projection algorithm (Flin et al., 2005, 2011). Figure 10 summarizes the results of this analysis. For all times, values are stable for subvolumes of size larger than 2 mm. So the REV of density (2.34 mm) can be chosen.

Figure 11 shows the evolution of SSA during the experiment. Starting at $(18 \pm 5\%)$ m$^2$ kg$^{-1}$, the SSA decreases slightly during the first 30 h and then increases more and more rapidly to reach about $(20 \pm 5\%)$ m$^2$ kg$^{-1}$.

### 3.4 Orientation tensor

Given the crystallographic orientation of each ice voxel, we can compute the second order orientation tensor (Fisher et al., 1993; Riche et al., 2013):

$$\mathbf{a^{(2)}} = \frac{1}{N_i} \sum_{i=1}^{N_i} \mathbf{c^i} \otimes \mathbf{c^i} \tag{1}$$

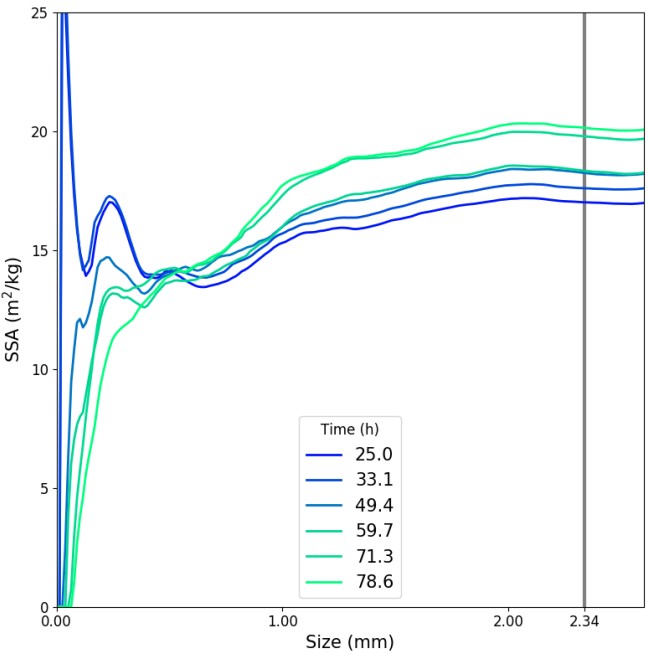

**Figure 10.** Specific Surface Area: REV estimation for different times. The vertical grey line represents the size of the REV volume taken for the remainder of the study.

with $c^i$ being the c-axis orientation of the i-th voxel in the volume, $\otimes$ the tensor product and $N_i$ the number of ice voxels. The orientation distribution is inscribed in an ellipsoïd whose principal axes are given by the eigenvectors $(\mathbf{v_1}, \mathbf{v_2}, \mathbf{v_3})$ of $\mathbf{a^{(2)}}$, and the length of the principal axis by its eigenvalues. Denoting the eigenvalues as $a_1$, $a_2$, $a_3$ with $1 > a_1 \geq a_2 \geq a_3 > 0$, we can identify 3 limit cases, defining 3 references of fabric.

– $a_1 > a_2 \simeq a_3$: cluster-type fabric, majority of c axis in one direction (defined by $\mathbf{v_1}$),

   – $a_1 \simeq a_2 > a_3$: girdle-type fabric, majority of c-axis in a plane (defined by $\mathbf{v_1}$ and $\mathbf{v_2}$),

   – $a_1 = a_2 = a_3 = 1/3$: isotropic fabric, same distribution in the three directions.

In order to evaluate the proximity of the measured distribution to those types, cluster index, girdle index and strength can be defined by $\ln(a_1/a_2)$, $\ln(a_2/a_3)$ and $\ln(a_1/a_3)$ respectively. The strength index equals 0 for a perfectly isotropic fabric. Figure
12 represents eigenvectors on the left panel, in pole figure representation, while the associated eigenvalues for the DCT images against time are shown on the right hand side. Eigenvalues lie in the range 0.279 to 0.405 without significant visible evolution over time. Eigenvectors are stable with time, one being parallel to temperature gradient, the two others being contained in the horizontal plane.

Figure 13 is a scatter plot of the cluster index against the girdle-index for the treated DCT volumes. The cluster index, girdle
index and strength index are respectively 0.188, 0.156 and 0.343 at the initial stage and 0.266, 0.083 and 0.349 at the final stage,





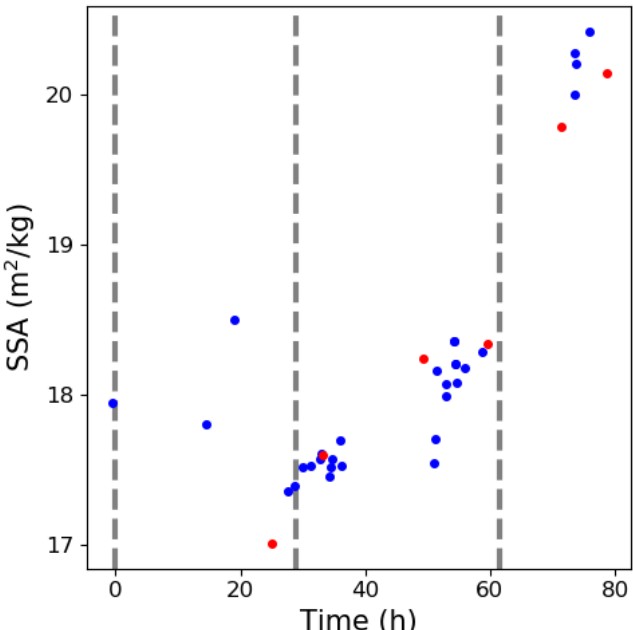

**Figure 11.** Evolution of the Specific Surface Area during the experiment. The grey vertical dashed lines represent the times of change of temperature conditions. Red dots correspond to times for which DCT information is available.

with mean values of 0.217, 0.117 and 0.335. Thus, the fabric is isotropic and stays isotropic during the whole experiment, while a trend to more cluster-like index can be observed.

### 3.5 Pore size distribution

Pore size distributions were computed on absorption images using a sphere fitting algorithm. The results are shown for several
5    times in Figure 14, with vertical lines denoting the mean of the corresponding distribution. In overall, the mean stays at about 190 µm and the distribution does not show any drastic evolution in shape. In particular, no trend is observable for the biggest pores, while a small trend is observable for the smaller ones. Indeed, the quantity of pores of size in the range 30 µm to 80 µm increases by about 1 %.

### 3.6 Relation between flux and grain orientation

10   The main objective of this study is to look at possible relations between local fluxes and crystalline orientation. In particular, we would like to test at the local scale the validity of orientation selective growth of grains under temperature gradient metamorphism.

From the 30 registered and segmented absorption images $(A_k)_{k \in [\![1,30]\!]}$, we computed local interface speeds $v_k$ where $k$ is the index of the absorption image. At a given point of the surface of the volume $k \in [\![1,30]\!]$ the normal $\mathbf{n_k}$ was computed using

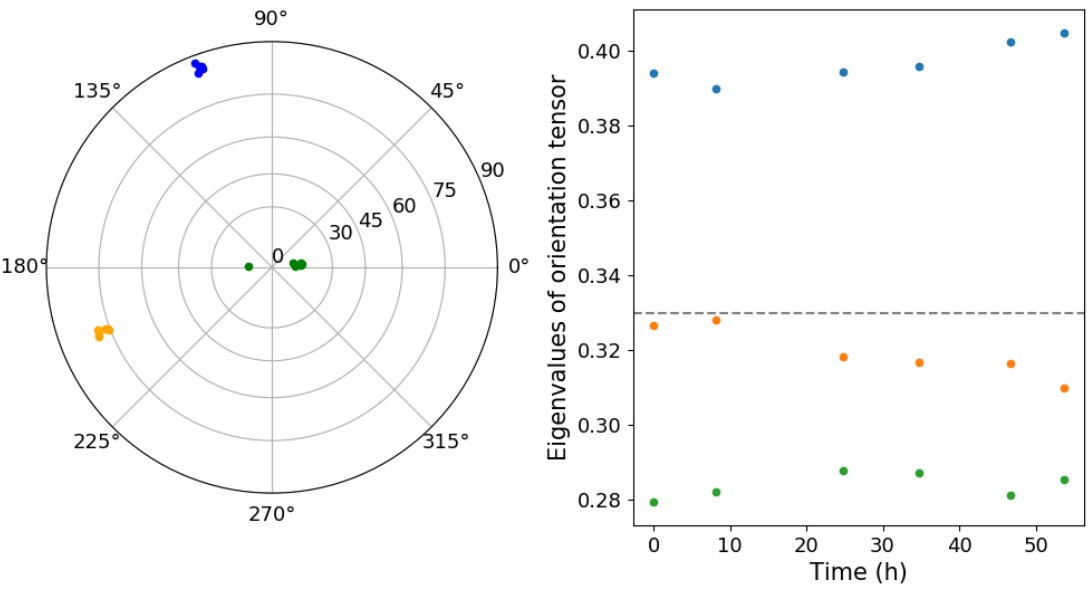

**Figure 12.** Evolution of the eigenvectors of the orientation tensors represented in pole figure with vertical axes as reference (left) and eigenvalues (right). Colours of dots for eigenvector on the left correspond to eigenvalue colours on the right.

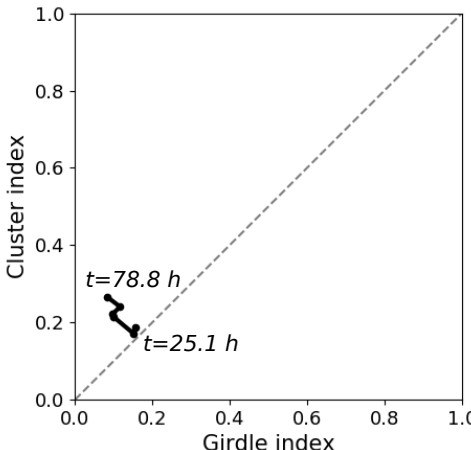

**Figure 13.** Cluster index versus girdle index for the different times indicated. Grey dashed line repesent the y=x line.

the ADGF algorithm (Flin et al., 2005). Noting $d_k^l$ the displacement of the interface along $\mathbf{n_k}$ from $A_k$ to $A_l$, we computed both $d_k^{k+1}$ and $d_k^{k-1}$ using a specific algorithm (Flin et al., 2018) whose principle is relatively close to the one presented by





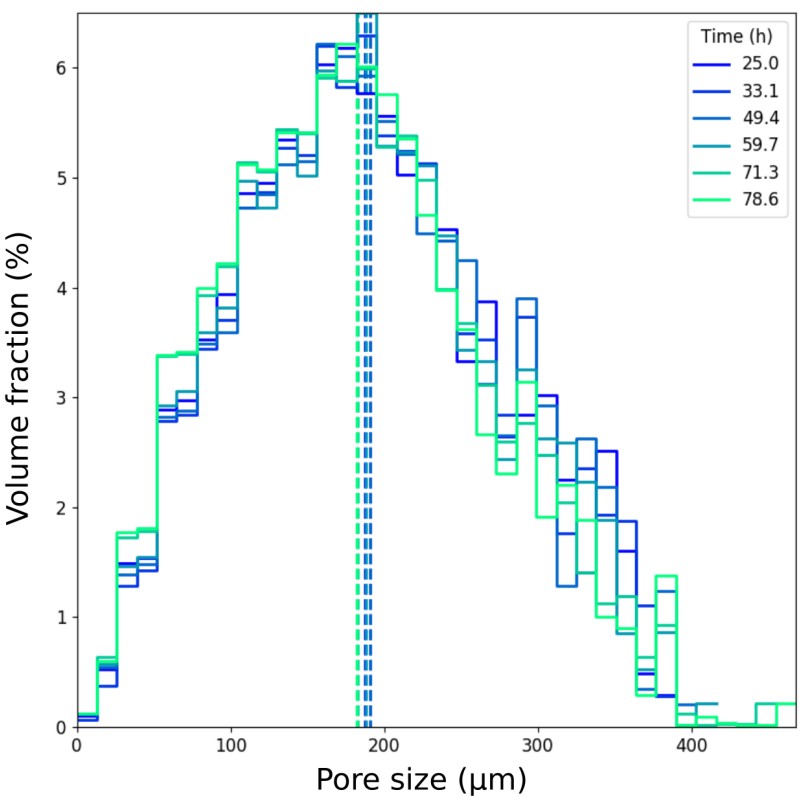

**Figure 14.** Pore size distribution, with 2 voxels = 13 mm bin size computed using a fitting sphere algorithm. Vertical dotted lines represent the mean of the associated distribution.

Krol and Löwe (2016). Then local normal velocity $v_k$ at the instant $t_k$ of acquisition of $A_k$, was estimated using :

$$v_k = \begin{cases} \dfrac{d_k^{k+1}}{t_{k+1} - t_k} & \text{if } k = 1 \\[2ex] \dfrac{t_k - t_{k-1}}{t_{k+1} - t_{k-1}} \dfrac{d_k^{k+1}}{t_{k+1} - t_k} - \dfrac{t_{k+1} - t_k}{t_{k+1} - t_{k-1}} \dfrac{d_k^{k-1}}{t_k - t_{k-1}} & \text{if } k \in [\![2, 29]\!] \\[2ex] -\dfrac{d_k^{k-1}}{t_k - t_{k-1}} & \text{if } k = 30 \end{cases} \qquad (2)$$

From the normal velocity of the interface, the mass flux $\phi_k$ of vapor changing into ice can be computed using:

$$\phi_k = \rho_i \cdot v_k \qquad (3)$$

5    Voxels were first classified depending on the values of $\alpha$ and $\gamma$ in 2D bins. Then, a mean flux value was computed for each bin. This enables us to represent a colourmap of the mean fluxes against $\alpha = (\mathbf{c}, \mathbf{n})$ and $\gamma = (\mathbf{z}, \mathbf{n})$ at the different times

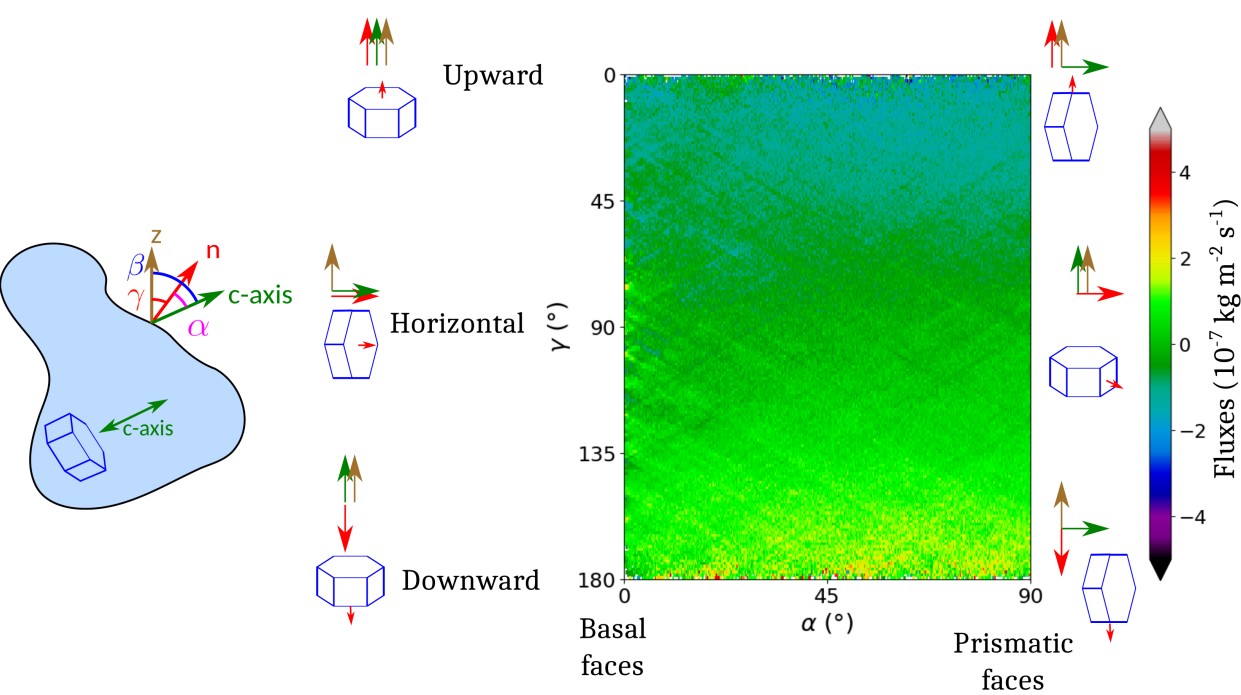

**Figure 15.** Interpretation of diagram of mean flux against $\alpha = (\mathbf{c}, \mathbf{n})$ and $\gamma = (\mathbf{z}, \mathbf{n})$. The small red arrows represent the facets taken into account.

of the experiment. An example of this representation is given in Figure 15. The pixels on the right of the diagram represent prismatic faces ($\alpha = 90°$) while the pixels on the left represent basal faces ($\alpha = 0$). The top diagram corresponds to surfaces pointing upward ($\gamma = 0$), while the bottom of the diagram represents surfaces pointing downward ($\gamma = 180°$); pixels at mid height of the diagram correspond to surfaces whose normals are horizontal ($\gamma = 90°$). A similar representation can be done by

5     representing the mean flux against $\beta = (\mathbf{z}, \mathbf{c})$ and $\gamma = (\mathbf{z}, \mathbf{n})$ (Figure 16). In that case, the right of the diagram represents grains with horizontal c-axis ($\beta = 90°$) while the left of the diagram corresponds to grains with vertical c-axis ($\beta = 0$).

     Representations for all times of mean flux against ($\alpha, \gamma$) are given in Figure 17, and against ($\beta, \gamma$) in Figure 18. At $t = 25.1$ h, from both figures, the absolute mean fluxes appear to be less than $10^{-8}$ kg m$^{-2}$ s$^{-1}$ in all cases. At $t = 33.3$ h, mean fluxes for $\gamma < 90°$ are negative (denoting sublimation) and positive (deposition) otherwise. This is due to the fact that the mean vapor

10     fluxes through the pore space are oriented upward: the faces whose normals point downward ($\gamma > 90°$) catch vapor which deposits while the faces oriented upward sublimate.

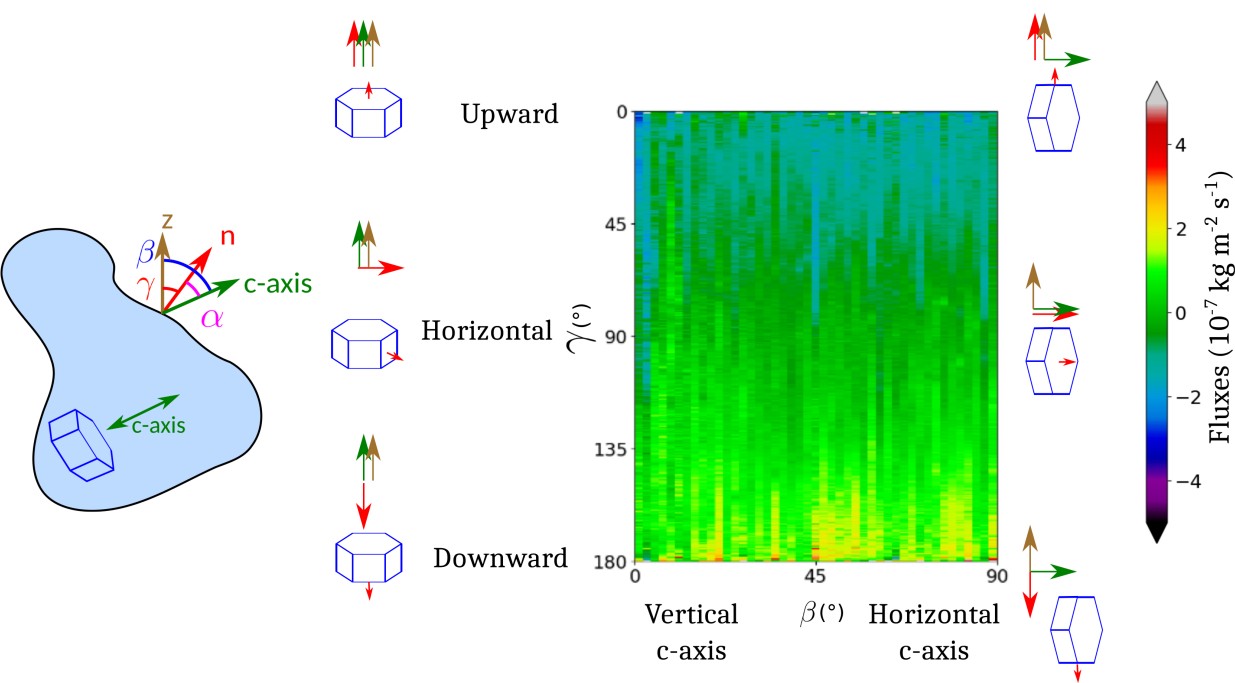

**Figure 16.** Interpretation of diagram of mean flux against $\beta = (\mathbf{z}, \mathbf{c})$ and $\gamma = (\mathbf{z}, \mathbf{n})$. The small red arrows represent the facets taken into account.

In addition, we can observe a second variation depending on the value of $\alpha$. Stronger rates of sublimation-deposition with absolute mean fluxes of $5 \times 10^{-7}$ kg m$^2$ s$^{-1}$ are found for $\alpha > 45°$. While there is no dependence on $\alpha$ for $\gamma = 90°$, the closer $\gamma$ gets to 0 or 180° the stronger is that dependence. Until $t = 78.8$ h, the same dependence on $\gamma$ is observed, and the dependence on $\alpha$ has the same shape for $\gamma < 90°$ while it is not visible for $\gamma > 90°$. Overall, at all points of the diagram, the values

5   are lower than at $t = 33.3$ h. The evolution between $t = 33.3$ h and $t = 78.8$ h is regular so that times $t = 49.9$ h, $t = 59.9$ h and $t = 71.8$ h are intermediate steps. The same observations can be made in Figure 18. The dependence on $\gamma$ differentiates between faces subject to deposition and those subject to sublimation. For $t \geq 33.3$ h, voxels belonging to crystals with a c-axis in the horizontal plane, or close to this plane ($\beta \in [45°, 90°]$) are more active.

## 3.7   Example of evolution inside a small subvolume

10   Figure 19 illustrates the various points observed from the data analysis. It shows the evolution of the microstructure inside a small cubic subsample of size 80 voxels = 520 μm. The colour of the surface refers to $\alpha$ so that prismatic and basal faces are represented in red and green, respectively. We can observe the formation of three faceted crystals, each with a different

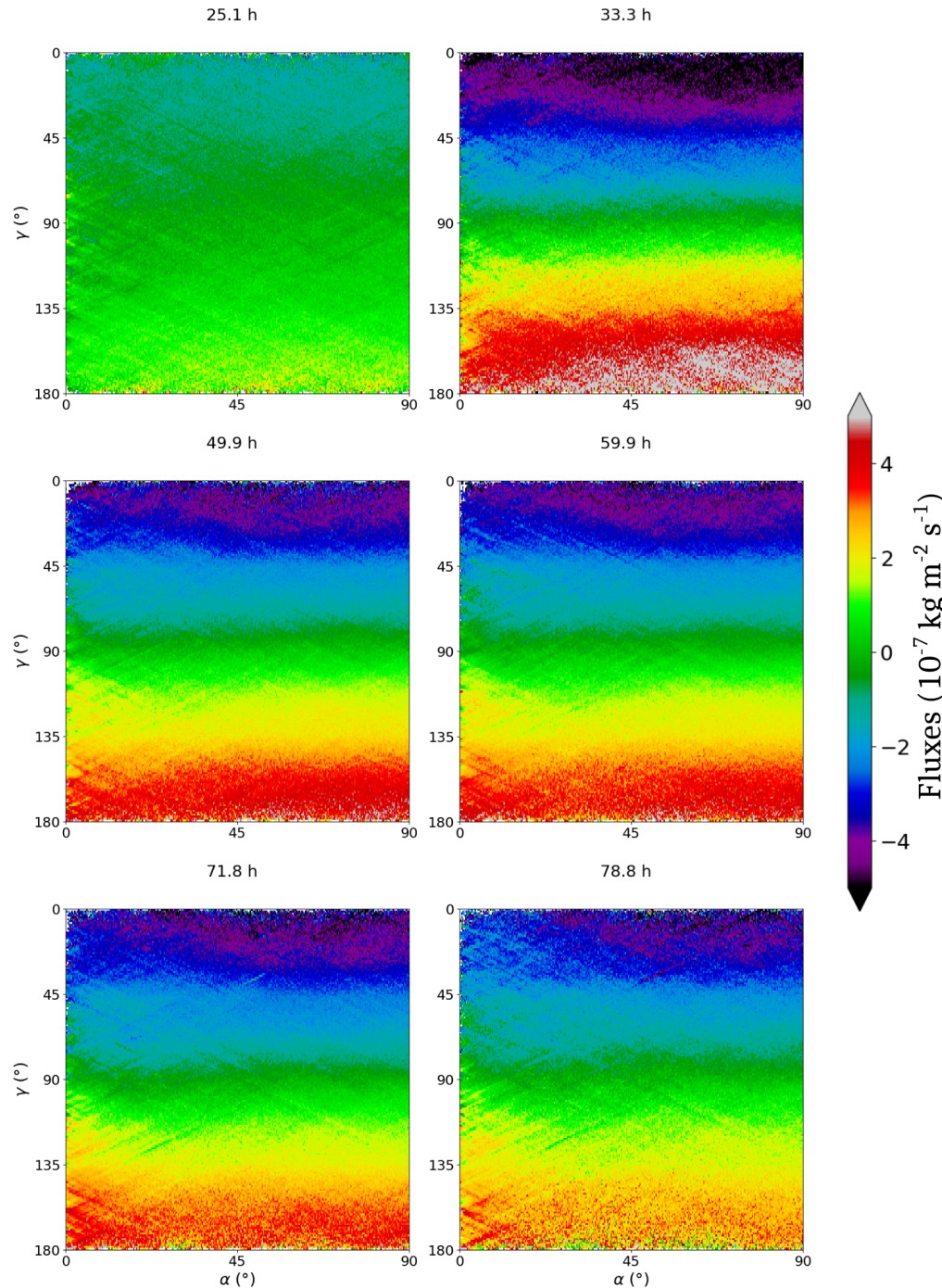

**Figure 17.** Mean fluxes against $\alpha = (\mathbf{c}, \mathbf{n})$ and $\gamma = (\mathbf{z}, \mathbf{n})$.





**Figure 18.** Mean fluxes against $\beta = (\mathbf{z}, \mathbf{c})$ and $\gamma = (\mathbf{z}, \mathbf{n})$.



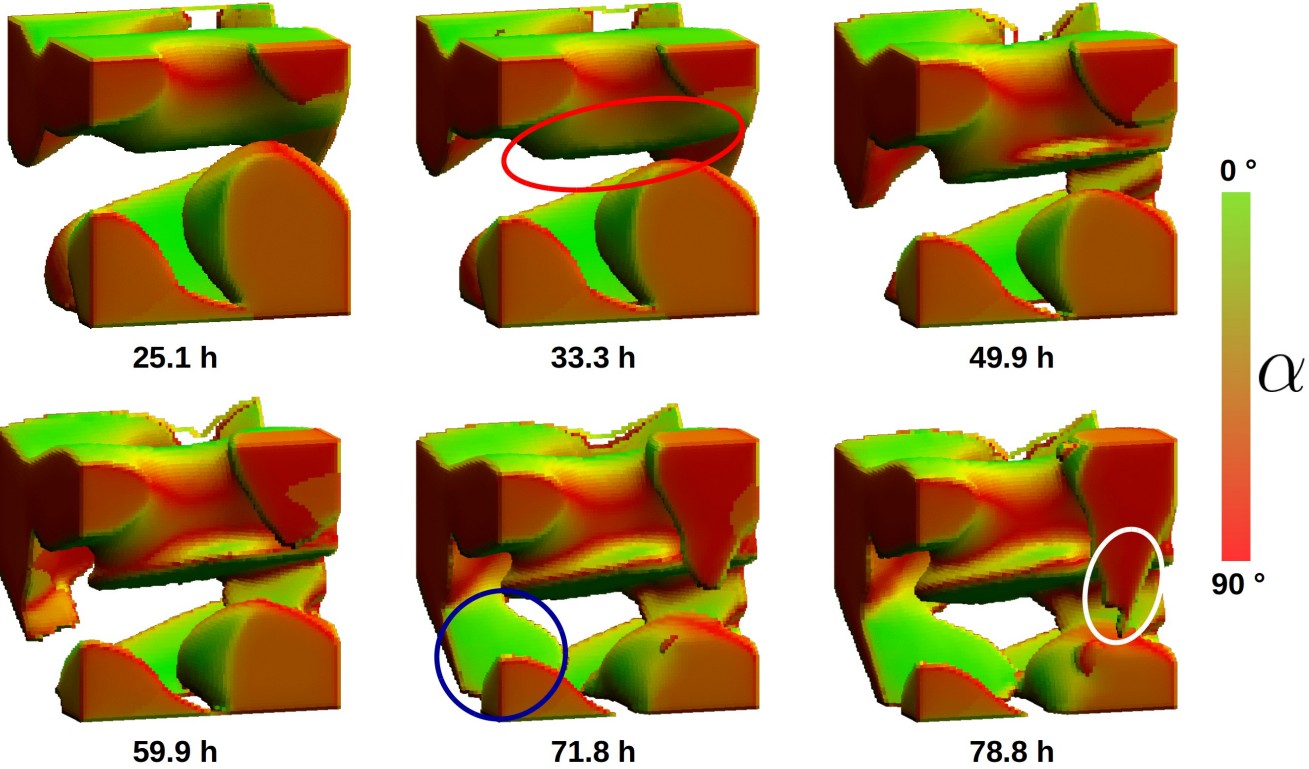

**Figure 19.** Example of evolution inside a small subvolume of 520 μm. Colours refer to the $\alpha$ value, with green representing basal faces and red prismatic faces.

orientation. The one pointed out by the red ellipse is almost horizontal while the one pointed out by the white ellipse is almost vertical. The blue localized facet has intermediate orientation.

The horizontal plate is slowly evolving while the one pointed out by the blue circle develops very well. Also, we can see some interactions between the plates pointed out by red and white ellipses. In particular, at $t = 78.8$ h, we observe that the

5 vertical plate grows at the expense of the horizontal one. Thus, while it is not a representative volume of the microstructure, this example illustrates well the observed trends in Figures 17 and 18.

## 4 Discussion

### 4.1 Fluxes and dependence on orientation

At $t = 25.1$ h, the fluxes and variations on fluxes are small. The temperature gradient of 17 °C m$^{-1}$ is the smallest used. We

10 conclude that in phase 1, metamorphism didn't act significantly due to small temperature gradient. This can be explained by




the relatively high density of the sample of $476\,\mathrm{kg\,m^{-3}}$. Indeed, it is known that high density snow requires higher temperature gradients (Akitaya, 1974; Marbouty, 1980; Pfeffer and Mrugala, 2002) and longer times of evolution to get visible modifications of the microstructure. Under a $52\,^\circ\mathrm{C\,m^{-1}}$ temperature gradient, the maximum values of the fluxes is of about $4\times10^{-7}$ $\mathrm{kg\,m^2\,s^{-1}}$. Pinzer et al. (2012) computed a macroscopic flux of about $3.5\times10^{-7}$ $\mathrm{kg\,m^2\,s^{-1}}$ for snow evolving under 49

$^\circ\mathrm{C\,m^{-1}}$ temperature gradient at -3.4 °C. Flin and Brzoska (2008) computed the highest fluxes of about $1.6\times10^{-7}\,\mathrm{kg\,m^2\,s^{-1}}$ under $16\,^\circ\mathrm{C\,m^{-1}}$. Thus, the present computed flux values are in very good agreement with this litterature.

   As described in subsection 3.6, for $t=33.3\,\mathrm{h}$, the most active points of the interface are for $\alpha{\geq}45^\circ$. We think that this corresponds to plates growing from grains, with prismatic faces being more active than basal faces. Indeed, from the definition of $\alpha$, the prismatic faces correspond to $\alpha=90^\circ$. Additionally, at the beginning of the experiment, the grain are typically

rounded so that all values of $\alpha$ are equally represented in terms of surface area. Since surfaces with $\alpha=45^\circ$, which have high index orientations, are rougher at the molecular level, they have higher deposition coefficient and are more active. Moreover, in this experiment, grown plates were really thin: about 2-3 voxels thick, as a consequence, normals computed on the side of these structures were inacurate, leading to several tens of degrees of uncertainty. These two effects explain why the highest fluxes are found in the range $\alpha\in45^\circ$ to $90^\circ$ and not just at $\alpha=90^\circ$. This observation is consistent with the mean temperature during

the experiment, which was always higher or equal to $-4^\circ$. Indeed, under that temperature condition, the kinetic coefficient is expected to be stronger on prismatic faces than on basal faces (Kuroda and Lacmann, 1982; Furukawa, 2015). Correspondingly, from Figure 18, the most active faces correspond to crystals for which $\beta=90^\circ$, i.e. with the c-axis in the horizontal plane. Following Adams and Miller (2003), this seems to confirm that kinetically favorably oriented crystals, in our case with basal faces oriented vertically and, thus, showing prismatic faces downward and upward, are indeed more active. At the end of the

experiment, the fluxes at the interface overall decrease. We see three reasons for that: firstly, the proportion of high index orientations (with $\alpha\simeq45^\circ$) disappear as the facets grow. Secondly, after the important growth of plates between $t=25\,\mathrm{h}$ and $t=33\,\mathrm{h}$, the growth, and the whole metamorphism, slows down because of geometric interactions between grains (see e.g. Figure 14, which shows the average pore size is slightly decreasing with time). Finally, at the end of the experiment, the mean temperature was reduced, thus reducing the sublimation-deposition rates.

**4.2   End of experiment: independence on orientation for deposition**

While a strong phase change is observed for grains with c-axis close to the horizontal plane at $t=33.3\,\mathrm{h}$, this progressively disappears on deposition surfaces. This suggest that at a certain point, growth intensity becomes less dictated by kinetics. We think that this may be explained by grain to grain interactions. At the beginning of phase 2, the microstructure has not evolved much, and a strong gradient triggers the growth of plates through the available pores. When the facets grow in the pore space

between several grains, growth locally occurs if the temperature of the grain is lower that the average temperature of the grain in interaction with that pore. But, as the facet grows through the pores, it becomes closer and closer to another grain, and growth becomes more dependent on the local configuration of the microstructure, and less in relation with the macroscopic temperature gradient. So, faceted grains develop well in pores and less efficiently after they have grown through the pores (see e.g. Figure 14). Similar conclusions on the necessity of pore space for growth have been drawn by Akitaya (1974). However,





on the sublimating surfaces, as the interface moves away from the pore, such a phenomenon does not occur, and fluxes stay well dependent on orientation.

### 4.3 Uncertainties due to processing

As explained in section 2.3, a dilation step is required at the end of the image processing route, in order to recover the orientation information missing due to dynamic diffraction. One can wonder if that could bias the Figures 17 and 18. When dilating the orientation information in one partially reconstructed grain, the correction is correct until the dilation front crosses a grain boundary. Beyond this, the correction is wrong. However, we assume the orientation difference between adjacent grains to be random. In consequence, that step affects the signal-to-noise ratio, which increases if the number of corrected voxels is higher than the number of errors. Errors are located in the vicinity of grain boundaries, and represent a very small fraction of the analyzed surface voxels and this implies no systematic error able to explain the observed trends. However, the improvement of the reconstruction assessing the problem of dynamic diffraction is of great interest, since it would increase the signal quality and permit other analyses. In particular, it seems crucial to properly study the grain boundaries. The problem of dynamic diffraction could be tackled by: i) improvement of diffraction spot thresholding and ii) modeling of the dynamic diffraction to reconstruct the grains from inhomogeneous intensities. The latter is a current subject of development of the technique.

### 4.4 Representativity

REV analysis for density and specific surface area shows that a 2.34 mm cubic subsample was representative of the volume sample as those quantities reach a plateau at that size. The dependence on $\gamma$ is consistent with a global vertical temperature gradient, suggesting that possible edge-effects due to the small size of the sample were negligible as compared to the effects of temperature gradient.

The density of 476 $\mathrm{kg\,m^{-3}}$ is high for dry snow, whose densities lie in the range 50 $\mathrm{kg\,m^{-3}}$ to 500 $\mathrm{kg\,m^{-3}}$. Akitaya (1974) noticed the occurrence of hard depth hoar for a snow density $> 300\,\mathrm{kg\,m^{-3}}$ exposed to temperature gradient of 39 $^\circ\mathrm{C\,m^{-1}}$. Pfeffer and Mrugala (2002) observed formation of hard depth hoar for snow of 400 $\mathrm{kg\,m^{-3}}$ for temperature gradient $> 20\,^\circ\mathrm{C\,m^{-1}}$ at a mean temperature of -7 $^\circ\mathrm{C}$ for 3 days. With a snow of density 230 $\mathrm{kg\,m^{-3}}$, they did not observe the formation of hard depth hoar, and concluded it exists a critical density between those values. In the present experiment, the transition occurred in phases 2 and 3 where the temperature gradient was 52 $^\circ\mathrm{C\,m^{-1}}$ with a mean temperature of about -2 $^\circ\mathrm{C}$ and -4 $^\circ\mathrm{C}$ respectively. So, results must be representative of the process of formation of hard depth hoar. In addition, both Riche et al. (2013) and Calonne et al. (2017) report no evolution of the fabric for the high-density snows, which would be consistent with the expectations that geometric-based competition between grains increases with density as the pore size decreases. However, the present results show that differences between grain orientations actually exist for growth and sublimation rates, even at high density. So this suggests that while the effects exist, they are less important in the selection of growing grains at that density and are more likely to occur at lower density.





## 5   Conclusions

For the first time, DCT monitoring of a temperature gradient metamorphism experiment was carried out. A snow sample of rounded grains with initial density of $(476 \pm 5\%)\ \mathrm{kg\,m^{-3}}$ has been subject to a temperature gradient of $17\ ^\circ\mathrm{C\,m^{-1}}$ for 1 day then to a temperature gradient of $52\ ^\circ\mathrm{C\,m^{-1}}$ for 2 days at a mean temperature of about -2°C for 2 days, and -4°C for 1 day.

Several points may be concluded from this experiment:

- The crystalline orientations have successfully been measured using DCT on about 900 growing and faceting grains.

- The local flux has been computed at each point of the interface, and the obtained value is well consistent with previous studies.

- Analyses show that the first mass transfers were overall higher on grains with the c-axis close to the horizontal plane, for both sublimation and deposition, thus confirming the hypothesis of orientation selective metamorphism, even at high density.

- Fluxes become independent of orientation at longer times for deposition while differences remain for sublimation. We attribute this effect to grain to grain interactions, particularly present at that density, preventing further growth of crystals.

Additionally, this study also opens new perspectives. Better accuracy on orientation, particularly at grains boundaries, can
be obtained by a proper management of dynamic diffraction, that occurs in ice, in the reconstruction process. Additionally, the observation of orientation selective grain growth and sublimation at high density gives confidence in the fact that this phenomenon actually occurs in more general conditions. The present experiment could be repeated in various conditions to characterize this effect in more detail. Finally, such data are necessary for models that describe the snow metamorphism using the crystalline nature of the ice.

**Acknowledgement**

We are specifically grateful to S. Rolland du Roscoat, F. Lahoucine, L. Pézard, J. Roulle, E. Boller, J-P. Valade and P. Tafforeau for their contributions to the set up of the experiment. We also thank A. Philip, P. Lapalus, A. Burr, A. Dufour, I. Peinke and P. Hagenmuller for their participation to the experiment monitoring.

We also would like to thank the ID19 beamline of the ESRF, where the images have been acquired.

This research as been partially founded by the MiMESis-3D ANR project (ANR-19-CE01-0009). CNRM/CEN is part of the LabEx OSUG@2020 (Investissements d'Avenir – grant agreement ANR10 LABX56). 3SR Laboratory is part of the LabEx Tec 21 (Investissements d'Avenir – grant agreement ANR11 269 LABX0030).



**Authors contributions**

R. Granger, and F. Flin developed the cryogenic cell. R. Granger, F. Flin, W. Ludwig and C. Geindreau, carried out the experiment. W. Ludwig and I. Hammad, post processed the DCT volumes. R. Granger, F. Flin and C. Geindreau analysed the data and wrote the article.

**Competing interests**

The authors declare that they have no conflict of interest.



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
