# Peer review of "Orientation selective grain sublimation-deposition in snow under temperature gradient metamorphism observed with Diffraction Contrast Tomography"

_The Cryosphere, 2021_

## Referee Comment (RC2)

**Review: "Orientation selective grain sublimation-deposition in snow under temperature gradient metamorphism observed with Diffraction Contrast Tomography" by Granger et al**

**General**

This paper investigates for the first time temperature gradient metamorphism by combined in-situ diffraction contrast and absorption tomography for a combined analysis of the coupled geometrical and crystallographic microstructure evolution in snow.

Seeing the feasibility of these kind of experiments is very exciting. This is without doubt very original work that warrants publication. Suggestions for improvements are given below including a few questions about the analysis from which the (in fact far-reaching) conclusion of "measurable signatures of crystal anisotropy in the interface evolution in TG metamorphism" is drawn.

And btw, apologies for the delayed review.

Best wishes,
Henning Löwe

**Specific**

(p1, l6): why "indeed"?

(p2, l2): "ensuring..." statement not clear.

(p3, l3): What are the specifications of CellDyM2 in terms of possible temperatures and temperature gradients when used in the beamline? Related to the question if the choice of experimental conditions used here is motivated technically or physically.

(p3, l7): "parasitic" does not fit here (and throughout)

(p4, l13): "metamorphised" $\rightarrow$ metamorphosed

(p5, l5): "consists in" $\rightarrow$ consists of

(p5, l8): what determines the conditions? why is COMSOL required?

(p5, Sec 2.2.): Is it possible to rule from the settings that that the absorbed energy is not sufficient to change the thermodynamic state of the snow sample? I just heard from other synchrotron experiments that this may be delicate.

(p13, l9): I don't understand, *all* indices are zero in an isotropic system, right?

(p14, l1): Given these values, I don't understand the statement that the fabric is isotropic.

(p15, l2): "Noting" $\rightarrow$ denoting by

(p15, l2): I couldn't find the paper (Flin 2018), the bibitem is incomplete. It's important to have sufficient information at hand on how $d$ is computed since it is essential for the conclusion.

(p14, l14): $k$ should be italic in $\boldsymbol{n}_k$

(p16, l5): What's the bin-size in angle space?

(p17, l8): "absolute mean fluxes" $\rightarrow$ mean absolute fluxes? What exactly is computed to be equal to $10^{-8}$?

(Figs 17,18): What causes the apparent "layering" in $\gamma$ direction? An artefact of the finite number of normal vector classes of the algorithm on voxel data? Hm, seems unlikely though, since the early stage doesn't show this layering. An explanation and a discussion of the relevance/irrelevance of this effect for the main conclusion seems justified.

(p18, l2-8): It's difficult to comprehend the statements about the $\alpha$ dependence as a function of $\gamma$ from the 2D histograms (It's nice though, that they are included). It would be helpful to re-cast these statements drawn from Fig 17,18 into one or two additional normal figures showing the $\alpha$ and $\beta$ dependence for $\gamma = 0, 90, 180$ with error bars for e.g. one example for each of the 3 experimental phases. This will also help in the discussion.

Along these lines of error bars: Since $\gamma = 0, 180$ corresponds to up/down interfaces with the largest displacements, the computation of normal distances may (depending on how its computed) be subject to the largest errors bars. So the question is how robust is the $\alpha, \beta$ signal for these orientations?

(Fig 17/18): Another question that comes to my mind: As described in Sec 3.6, mean fluxes computed form Eq. (3) are used to assess the orientation dependence of the growth law via the dependence on $\alpha, \beta$ angle coordinates. As far as I understand, angles were determined *per surface node* of the triangulated interface mesh and the flux via Eq. (2). Can it be ruled out that there is no artificial correlation left of the "flux" on the underlying mesh heterogeneity? Surface patches in facet regions will likely end up with larger triangles than high-index surfaces due to the difference in curvatures. Doesn't a computation of fluxes from nodal $\alpha, \beta$ data without including area-weights understimate the fluxes on facets?

(p21, l3): What do you mean by "slowly evolving" and "evolving well"? It might be more concise to rephrase these statements in terms of the angles and velocities introduced before. This also makes the link of the "small cutout example" to the histograms shown before more tight.

(p22, l3): Is this really the total maximum flux/velocity computed? Or is it the maximum over the bin-mean? Then it depends on bin size?

(p22, l6): "litterature" $\rightarrow$ literature

(p22, l16): "stronger" $\rightarrow$ higher

(Discussion general): One aspect that should be included in the discussion is the common observation of an up/down asymmetry between sublimating and growing surfaces in TG metamorphism (e.g. Calonne TC, 2014 / Krol, Acta, 2018) How does this align with the present interpretation?

(p22, l17): I still don't entirely understand how the effect of "kinetic favor" ($\beta = 90$) is eventually discerned in the data from the effect of being "geometrically favorable" ($\gamma = 0, 180$) with respect to the TG direction. This connects back to the comment above (p18, l2-8), where a remaining $\beta$ signal (beyond error bars) should be detectable for having ($\gamma = 0, 180$) fixed.

(p22, Sec 4.2): Also the statements in this section would benefit from an additional contraction (as suggested above) of the information in the 2D histograms into additional normal figures.

---

## Author Comment (AC2)

(p1, l6): why "indeed"?'Indeed' has been removed.(p2, l2): "ensuring..." statement not clear.'ensuring' has been replaced by 'leading to'.

(p3, l3): What are the specifications of CellDyM2 in terms of possible temperatures and temperature gradients when used in the beamline? Related to the question if the choice of experimental conditions used here is motivated technically or physically.

The cell gives the possibility to precisely impose the temperature (about -40 to +40 °C) at the top and bottom of the sample, ensuring a large variety of temperature gradients. So the cell did not imply any specific constraint on the conditions of the experiment. Conditions have been chosen in order to have the right speed of evolution, i.e. a speed i) large enough to observe evolution during the allocated beamtime but ii) not too fast to be able to track the local evolution of each grain. In addition, the choice of not using fresh snow but snow stored for 7 months at stable temperature was made in order to avoid image resolution and grain rearrangement issues, allowing us to focus on water vapor flow.

(p3, 17): "parasitic" does not fit here (and throughout) Parasitic seems adapted in that case. (p4, 113): "metamorphised"  $\rightarrow$  metamorphosed Changed. (p5, 15): "consists in"  $\rightarrow$  consists of Changed.

(p5, l8): what determines the conditions? why is COMSOL required?

As explained in Calonne et al., (2015), the Peltier modules control the temperature at the ends of the copper columns, snow sample being at the other ends. The simulation permits a refined estimation of the actual temperature at the snow sample boundaries, taking into account several parameters such as the properties of the snow sample, conduction of other involved materials, vaccuum level of the chamber, room temperature, etc.

(p5, Sec 2.2.): Is it possible to rule from the settings that the absorbed energy is not sufficient to change the thermodynamic state of the snow sample? I just heard from other synchrotron experiments that this may be delicate.

The X rays were turned off between the scans. If one looks to quantities coming from the Peltier modules, namely the current through the Peltier modules and the cold temperatures against time (figure below), it can be seen that the Peltier modules were not using more power to maintain the temperature at the target at the scans times (black vertical dashed lines) than for the rest of the time. This suggests that the heat source due to X rays absorption by the sample was not strong enough to modify its thermodynamic state.

(p13, l9): I don't understand, all indices are zero in an isotropic system, right? Changed.

(p14, l1): Given these values, I don't understand the statement that the fabric is isotropic. Statement reformulated.

(p15, l2): "Noting"  $\rightarrow$  denoting by **Changed.**

(p15, l2): I couldn't find the paper (Flin 2018), the bibitemis incomplete. It's important to have sufficient information at hand on how d is computed since it is essential for the conclusion.

**The bibitem has been corrected an completed by:**

Flin, F., Calonne, N., Lesaffre, B., Dufour, A., Philip, A., Rolland du Roscoat, S., and Geindreau, C. (2014). The TG metamorphism of snow: Toward an evaluation of 3D numerical models by timelapse images acquired under X-ray tomography ? Physics and Chemistry of Ice (PCI), Hanover, USA, 17 - 20 March, 2014.

Flin, F., Denis, R., Mehu, C., Calonne, N., Lesaffre, B., Dufour, A., Granger, R., Lapalus, P., Hagenmuller, P., Roulle, J., Rolland du Roscoat, S., Bretin, E., and Geindreau, C. (2018). Isothermal metamorphism of snow: Measurement of interface velocities and phase-field modeling for a better understanding of the involved mechanisms. Physics and Chemistry of Ice (PCI), Zuerich, Switzerland, 8 - 12 January, 2018.

(p14, l14): k should be italic in nk

Changed.

(p16, l5): What's the bin-size in angle space? Bin-sizes are the following:

gamma: 1°

alpha: 0.25°

beta: 2°

Bin size have been optimized to be the smallest as possible, but having sufficiently data in each of them.

(p17, l8): "absolute mean fluxes"  $\rightarrow$  mean absolute fluxes? What exactly is computed to be equal to 10–8?

**This statement has been reformulated.**

(Figs 17,18): What causes the apparent "layering" in  $\gamma$  direction? An artefact of the finite number

of normal vector classes of the algorithm on voxel data? Hm, seems unlikely though, since the early stage doesn't show this layering. An explanation and a discussion of the relevance/irrelevance of this effect for the main conclusion seems justified.

The colored horizontal bands are due to the fact that the more the surface normal is oriented against the flux (vertical upward), the more it catch fluxes, with similar effect for sublimation when the normal is oriented upward. This has been commented at p 17, l 10.

If « layering » refers to apparent vertical lines in figure 18, this comes from the fact that  $\beta$  is the orientation of the grains, i.e. there is only one  $\beta$  per grain. This means there is not as many  $\beta$  values in the dataset than for  $\alpha$  and  $\gamma$  as those are computed at each interface voxel location. In consequence, the bin size is larger for  $\beta$ , and data are noisier, leading to those vertical lines.

(p18, l2-8): It's difficult to comprehend the statements about the  $\alpha$  dependence as a function of  $\gamma$  from the 2D histograms (It's nice though, that they are included). It would be helpful to re-cast these statements drawnfrom Fig 17,18 into one or two additional normal figures showing the  $\alpha$  and  $\beta$  dependence for  $\gamma$  =0,90, 180 with error bars for e.g. one example for each of the 3 experimental phases. This will also help in the discussion.

Along these lines of error bars: Since  $\gamma = 0$ , 180 corresponds to up/down interfaces with the largest displacements, the computation of normal distances may (depending on howits computed) be subject to the largest errors bars. So the question is how robust is the  $\alpha$ , $\beta$  signal for these orientations?

---

## Author Response (AR1)

**Answer to H. Loewe's comment:**

(p1, l6): why "indeed"?
'Indeed' has been removed.
(p2, l2): "ensuring..." statement not clear.
'ensuring' has been replaced by 'leading to'.

(p3, l3): What are the specifications of CellDyM2 in terms of possible temperatures and temperature gradients when used in the beamline? Related to the question if the choice of experimental conditions used here is motivated technically or physically.
The cell gives the possibility to precisely impose the temperature (about -40 to +40 °C) at the top and bottom of the sample, ensuring a large variety of temperature gradients. So the cell did not imply any specific constraint on the conditions of the experiment. Conditions have been chosen in order to have the right speed of evolution, i.e. a speed i) large enough to observe evolution during the allocated beamtime but ii) not too fast to be able to track the local evolution of each grain. In addition, the choice of not using fresh snow but snow stored for 7 months at stable temperature was made in order to avoid image resolution and grain rearrangement issues, allowing us to focus on water vapor flow.

(p3, l7): "parasitic" does not fit here (and throughout)
Parasitic seems adapted in that case.
(p4, l13): "metamorphised" → metamorphosed
Changed.
(p5, l5): "consists in" → consists of
Changed.

(p5, l8): what determines the conditions? why is COMSOL required?
As explained in Calonne et al., (2015), the Peltier modules control the temperature at the ends of the copper columns, snow sample being at the other ends. The simulation permits a refined estimation of the actual temperature at the snow sample boundaries, taking into account several parameters such as the properties of the snow sample, conduction of other involved materials, vaccuum level of the chamber, room temperature, etc.

(p5, Sec 2.2.): Is it possible to rule from the settings that the absorbed energy is not sufficient to change the thermodynamic state of the snow sample? I just heard from other synchrotron experiments that this may be delicate.
The X rays were turned off between the scans. If one looks to quantities coming from the Peltier modules, namely the current through the Peltier modules and the cold temperatures against time (figure below), it can be seen that the Peltier modules were not using more power to maintain the temperature at the target at the scans times (black vertical dashed lines) than for the rest of the time. This suggests that the heat source due to X rays absorption by the sample was not strong enough to modify its thermodynamic state.

[Figure]

[Figure]

(p13, l9): I don't understand, all indices are zero in an isotropic system, right?
Changed.
(p14, l1): Given these values, I don't understand the statement that the fabric is isotropic.
Statement reformulated.
(p15, l2): "Noting" → denoting by
Changed.

(p15, l2): I couldn't find the paper (Flin 2018), the bibitemis incomplete. It's important to have sufficient information at hand on how d is computed since it is essential for the conclusion.

The bibitem has been corrected an completed by:
Flin, F., Calonne, N., Lesaffre, B., Dufour, A., Philip, A., Rolland du Roscoat, S., and Geindreau, C. (2014). The TG metamorphism of snow: Toward an evaluation of 3D numerical models by time-lapse images acquired under X-ray tomography ? Physics and Chemistry of Ice (PCI), Hanover, USA, 17 - 20 March, 2014.
Flin, F., Denis, R., Mehu, C., Calonne, N., Lesaffre, B., Dufour, A., Granger, R., Lapalus, P., Hagenmuller, P., Roulle, J., Rolland du Roscoat, S., Bretin, E., and Geindreau, C. (2018). Isothermal metamorphism of snow: Measurement of interface velocities and phase-field modeling for a better understanding of the involved mechanisms. Physics and Chemistry of Ice (PCI), Zuerich, Switzerland, 8 - 12 January, 2018.

(p14, l14): k should be italic in nk
Changed.
(p16, l5): What's the bin-size in angle space?
Bin-sizes are the following:
gamma: 1°
alpha: 0.25°
beta: 2°
Bin size have been optimized to be the smallest as possible, but having sufficiently data in each of them.

(p17, l8): "absolute mean fluxes" → mean absolute fluxes? What exactly is computed to be equal to $10^{-8}$?
This statement has been reformulated.

(Figs 17,18): What causes the apparent "layering" in γ direction? An artefact of the finite number

of normal vector classes of the algorithm on voxel data? Hm, seems unlikely though, since the early stage doesn't show this layering. An explanation and a discussion of the relevance/irrelevance of this effect for the main conclusion seems justified.

The colored horizontal bands are due to the fact that the more the surface normal is oriented against the flux (vertical upward), the more it catch fluxes, with similar effect for sublimation when the normal is oriented upward. This has been commented at p 17, l 10.

If « layering » refers to apparent vertical lines in figure 18, this comes from the fact that β is the orientation of the grains,  i.e. there is only one β per grain. This means there is not as many β values in the dataset than for α and γ as those are computed at each interface voxel location. In consequence, the bin size is larger for  β, and data are noisier, leading to those vertical lines.

(p18, l2-8): It's difficult to comprehend the statements about the α dependence as a function of γ fromthe 2Dhistograms (It's nice though, that they are included). It would be helpful to re-cast these statements drawnfromFig 17,18 into one or two additional normal figures showing the αand β dependence for γ =0,90, 180 with error bars for e.g. one example for each of the 3 experimental phases. This will also help in the discussion.

Along these lines of error bars: Since γ = 0, 180 corresponds to up/down interfaces with the largest displacements, the computation of normal distances may (depending on howits computed) be subject to the largest errors bars. So the question is how robust is the α,β signal for these orientations?

[Figure]

We added an additional figure showing the flux dependence upon α,γ: for each class of α, it displays the mean value of the bin-averaged flux (i.e. mean of the values displayed by Fig. 17) for γ in the range 0-20° (blue), 80-100° (green) and 160-180° (red). The shaded area corresponds to the associated uncertainties. Such uncertainties come from the measurement of the distance between two successive positions of the interface, which is estimated to be 1 voxel, i.e. 6.5 µm.

At all times, for α varying from 0° to ~5 ° one can see a minor decrease of the fluxes for γ in [0°, 20°] and γ in [80°, 100°] and an increase for γ in [160°, 180°]. This corresponds to an artefact arising from the discretization of the angles with the regular grid imposed by the pixelated image, and is also visible in Fig. 17. It should be ignored for the rest of the interpretation.

At time 33.3 h, for α varying from 5° to 90 °, one can see a trend for the fluxes varying from ~4x10$^{-7}$ kg/m²/s to ~5x10$^{-7}$ kg/m²/s for γ in [160°, 180°], and from ~-4x10$^{-7}$ kg/m²/s to ~-5x10$^{-7}$ kg/m²/s for γ in [0°, 20°].

This suggests that among interfaces geometrically favored (γ in [0°,20°] for sublimation, γ in [160°, 180°] for condensation), the ones kinetically favored (α close to 90°), are more subject to phase change than the ones kinetically defavored (α close to 0°).

In the same way, at t = 78.8h, for γ in [0°, 20°], one can see a variation from -3x10⁻⁷ kg/m²/s to -4x10⁻⁷ kg/m²/s. While there is no variation anymore for γ in [0°, 20°].

One note also that at all times, for γ in [80°, 100°], the flux is ~0x10⁻⁷ kg/m²/s with no observable variations with α. This comes from the fact that vertical interfaces do not act strongly in the vertical mass transfer.

(Fig 17/18): Another question that comes to my mind: As described in Sec 3.6, mean fluxes computed formEq. (3) are used to assess the orientation dependence of the growth law via the dependence on α,β angle coordinates. As far as I understand, angles were determined per surface node of the triangulated interface mesh and the flux via Eq. (2). Can it be ruled out that there is no artificial correlation left of the "flux" on the underlying mesh heterogeneity? Surface patches in facet regions will likely end up with larger triangles than high-index surfaces due to the difference in curvatures. Doesn't a computation of fluxes from nodal α,β data without including area-weights understimate the fluxes on facets?

Conceptually, the mass flux is computed as follows:
The algebraic mass of ice appearing on an element of surface is computed from the volume of ice defined by i) the local surface element and ii) the normal distance between two successive images.

The flux corresponds to a mass per unit of time and unit of surface:  the surface to take into account here is the surface on which that mass actually deposits, i.e. the surface element which simplifies with i). The time to take into account is the time between images, combined with ii) it gives the local growth rate.

Practically, as a result one finds the equation (3) for computing the flux.

(p21, l3): What do you mean by "slowly evolving" and "evolving well"? It might be more concise to rephrase these statements in terms of the angles and velocities introduced before. This also makes the link of the "small cutout example" to the histograms shown before more tight.

This sentence was reformulated. We completed the Fig.19 with the visualization of the grain map, the α, β, γ angles and the interface fluxes on the microstructure.

[Figure]

(p22, l3): Is this really the total maximumflux/velocity computed? Or is it the maximumover
the bin-mean? Then it depends on bin size?
The statement as been reformulated: it is the maximum over the bin-mean (i.e. the maximum of the value
displayed in the figure).
Here we estimate the bin size to be sufficiently smaller than the variation in angle to get significant change on
the flux: as it can be seen on the figures, the red/purple colors clearly spread over a larger area than bin sizes.

(p22, l6): "litterature" → literature
Changed.
(p22, l16): "stronger" → higher
Changed.

(Discussion general): One aspect that should be included in the discussion is the common obser-
vation of an up/down asymmetry between sublimating and growing surfaces in TGmetamorphism
(e.g. Calonne TC, 2014 / Krol, Acta, 2018) How does this align with the present interpretation?

The common asymmetry between sublimating/condensing interfaces is geometric, and corresponds to the fact
that sublimating interfaces get rounded, while depositing surfaces get faceted. Here, the analysis does not
extract such geometrical information. Instead, we focused on the role of the crystalline orientation on the
deposition (resp. sublimation) rates. Of course, the present results are fully compatible with the up/down
assymetry described in other papers.

(p22, l17): I still don't entirely understand howthe effect of "kinetic favor" (β =90) is eventually
discerned in the data fromthe effect of being "geometrically favorable" (γ =0, 180) with respect
to the TGdirection. This connects back to the comment above (p18, l2-8), where a remaining β
signal (beyond error bars) should be detectable for having (γ =0, 180) fixed.
(p22, Sec 4.2): Also the statements inthis sectionwould benefit froman additional contraction (as
suggested above) of the information in the 2Dhistograms into additional normal figures.

See answers above.

**Answer to I. Baker's comment:**

Really excellent, ground-breaking paper.

Some minor comments:

1. Why not work on fresh snow rather than snow stored for 7 months. Perhaps, the authors can comment on this.

2. We are told the pore size increases 1%. How real is this? How big are the error bars?

3. "Representativity" should be changed to the real English word of "Representivity".

1. The objective was to measure local phase change fluxes against the orientation of crystals relatively to the temperature gradient direction. Working on fresh snow would have lead to possible image resolution issues as well as to a rearrangement of the grains, thus complexifying the following analysis. Working on snow stored at -20°C for 7 months permits to overcome these problems. This was clarified in the text.

2. The sentence in the original version of the article is indeed confusing: the quantity of pores with sizes between 30 µm and 80 µm increases from 3 % to about 4 % of the total pore volume. Thus, the proportion in that range of sizes increases by 33 % of its initial value. This was clarified in the text.
Concerning the error bars: errors in the pore size measurement at a point of the microstructure would arise from segmentation and discretisation of the microstructure, and might lead to an error of the pore size of about 2 voxels = 13 µm. The effect of this error on the distribution is a random misclassifaction of each point of the pore space, over a range with length 13 µm centered on its actual value. In other words, this error leads to a smoothing of the pore size distribution on a window of about 13 µm. Here, the range over which the trend is observed (> 50 µm) is significantly larger than the estimated errors above. Thus, we are rather confident with the fact that the observed trend actually exists.
We can also notice that this increase of small pores is consistent with the increase of SSA observed in Fig. 11.

3. This has been corrected in the new version of the article.

---

## Author Response (AR2)

Dear editor and referees,

Many thanks for your final revision and the acceptance of our manuscript.
L 201, we meant 476 kg m-3 (+/- 5%), this has been modified in the text, and we made the technical corrections you proposed.

Best Regards